# A convenient online desalination tube coupled with mass spectrometry for the direct detection of iodinated contrast media in untreated human spent hemodialysates

Md. Mahamodun Nabi[1,2], Takumi Sakamoto[1], Md. Al Mamun[1], Ariful Islam[1], A. S. M. Waliullah[1], Shuhei Aramaki[1,3], Md. Mahmudul Hasan[1], Shingo Ema[4], Akihiko Kato[4,5], Yutaka Takahashi[1,6], Tomoaki Kahyo[1,7], Mitsutoshi Setou[1,6,7,8], Tomohito Sato[1,6,7,9]*

1 Department of Cellular & Molecular Anatomy, Hamamatsu University School of Medicine, Hamamatsu, Shizuoka, Japan, 2 Institute of Food & Radiation Biology, Atomic Energy Research Establishment, Bangladesh Atomic Energy Commission, Savar, Dhaka, Bangladesh, 3 Department of Radiation Oncology, Hamamatsu University School of Medicine, Hamamatsu, Shizuoka, Japan, 4 Blood Purification Therapy Department, Medical Device Management Department Hamamatsu University Hospital, Hamamatsu, Shizuoka, Japan, 5 First Department of Medicine, Hamamatsu University School of Medicine, Hamamatsu, Shizuoka, Japan, 6 Preppers Co. Ltd., Hamamatsu University School of Medicine, Hamamatsu, Shizuoka, Japan, 7 International Mass Imaging Center, Hamamatsu University School of Medicine, Hamamatsu, Shizuoka, Japan, 8 Department of Systems Molecular Anatomy, Institute for Medical Photonics Research, Preeminent Medical Photonics Education & Research Center, Hamamatsu, Shizuoka, Japan, 9 First Department of Surgery, Hamamatsu University School of Medicine, Hamamatsu, Shizuoka, Japan

* tsato@hama-med.ac.jp

**Data Availability Statement:** Raw data files are available from the Biostudies database. https://www.ebi.ac.uk/biostudies/ Accession: S-BSST844.

## Abstract

### Background

Mass spectrometry (MS) analysis using direct infusion of biological fluids is often problematic due to high salts/buffers. Iodinated contrast media (ICM) are frequently used for diagnostic imaging purposes, sometimes inducing acute kidney injury (AKI) in patients with reduced kidney function. Therefore, detection of ICM in spent hemodialysates is important for AKI patients who require urgent continuous hemodiafiltration (CHDF) because it allows noninvasive assessment of the patient's treatment. In this study, we used a novel desalination tube before MS to inject the sample directly and detect ICM.

### Methods

Firstly, spent hemodialysates of one patient were injected directly into the electrospray ionization (ESI) source equipped with a quadrupole time-of-flight mass spectrometer (Q-TOF MS) coupled to an online desalination tube for the detection of ICM and other metabolites. Thereafter, spent hemodialysates of two patients were injected directly into the ESI source equipped with a triple quadrupole mass spectrometer (TQ-MS) connected to that online desalination tube to confirm the detection of ICM.

**Funding:** This research work was supported by MEXT Project for promoting public utilization of advanced research infrastructure (Imaging Platform) (Grant number JPMXS0410300220), Japan.

**Competing interests:** The authors have declared that no competing interests exist.

## Results

We detected iohexol (an ICM) from untreated spent hemodialysates of the patient-administered iohexol for computed tomography using Q-TOF MS. Using MRM profile analysis, we have confirmed the detection of ICM in the untreated spent hemodialysates of the patients administered for coronary angiography before starting CHDF. Using the desalination tube, we observed approximately 178 times higher signal intensity and 8 times improved signal-to-noise ratio for ioversol (an ICM) compared to data obtained without the desalination tube. This system was capable of tracking the changes of ioversol in spent hemodialysates of AKI patients by measuring spent hemodialysates.

## Conclusion

The online desalination tube coupled with MS showed the capability of detecting iohexol and ioversol in spent hemodialysates without additional sample preparation or chromatographic separation. This approach also demonstrated the capacity to monitor the ioversol changes in patients' spent hemodialysates.

## Introduction

Mass spectrometry (MS) has emerged as a powerful analytical technique in various disciplines, including biomedical research, in recent years. The use of MS is increasing in biomedical science day by day due to its capability of high throughput screening of new uremic analytes [1–3] and toxins [4]. Among the various ionization methods, electrospray ionization (ESI) is widely used to study heterogeneous complex mixtures due to its high sensitivity [5]. To ensure a stable ESI process, the presence of excess salts and nonvolatile solutes/buffers in aqueous solutions is not desired [6]. A high concentration of salts, nonvolatile buffers, and endogenous metabolites (amines, fatty acids, etc.) in bio-fluids interfere with ESI-MS analysis significantly in many ways, such as orifice/interface contamination, clogging of ionization portion, ion suppression [7], and peak splitting [8]. However, some bio-fluids are salty (e.g., cerebrospinal fluids contain 145 mM Na) [9], while others need nonvolatile salts buffers and solubilizing agents (such as organic solvents or detergents) to preserve the stability and integrity of biomolecules and their complexes [5]. Therefore, direct analysis of untreated bio-fluids by MS represents a considerable challenge.

Hemodiafiltration (HDF) has become the key management modality for patients with various acute kidney injury (AKI) entities such as contrast media-induced AKI and nephropathy [10], septic AKI, and rhabdomyolysis-associated AKI [11]. It has emerged as an alternative to current innovative renal replacement therapy, which removes higher molecular weights uremic solutes more efficiently than conventional hemodialysis [12]. A significant amount of dialysis concentrate is required to maintain the homeostasis of bodily fluids, electrolytes, osmolality, pH, and the removal of toxic products from the bloodstream simultaneously. This procedure produces large quantities of spent hemodialysates (Fig 1A), containing much of the patient's information, such as metabolites removed from the body and drugs administered. It would be advantageous and noninvasive if we could use spent hemodialysates effluent to determine the patient's clinical condition. However, spent hemodialysates contain inorganic salts and nonvolatile solutes/buffers, making them more challenging to analyze by MS [1]. To address these challenges, a convenient desalination tube was installed just before the ESI

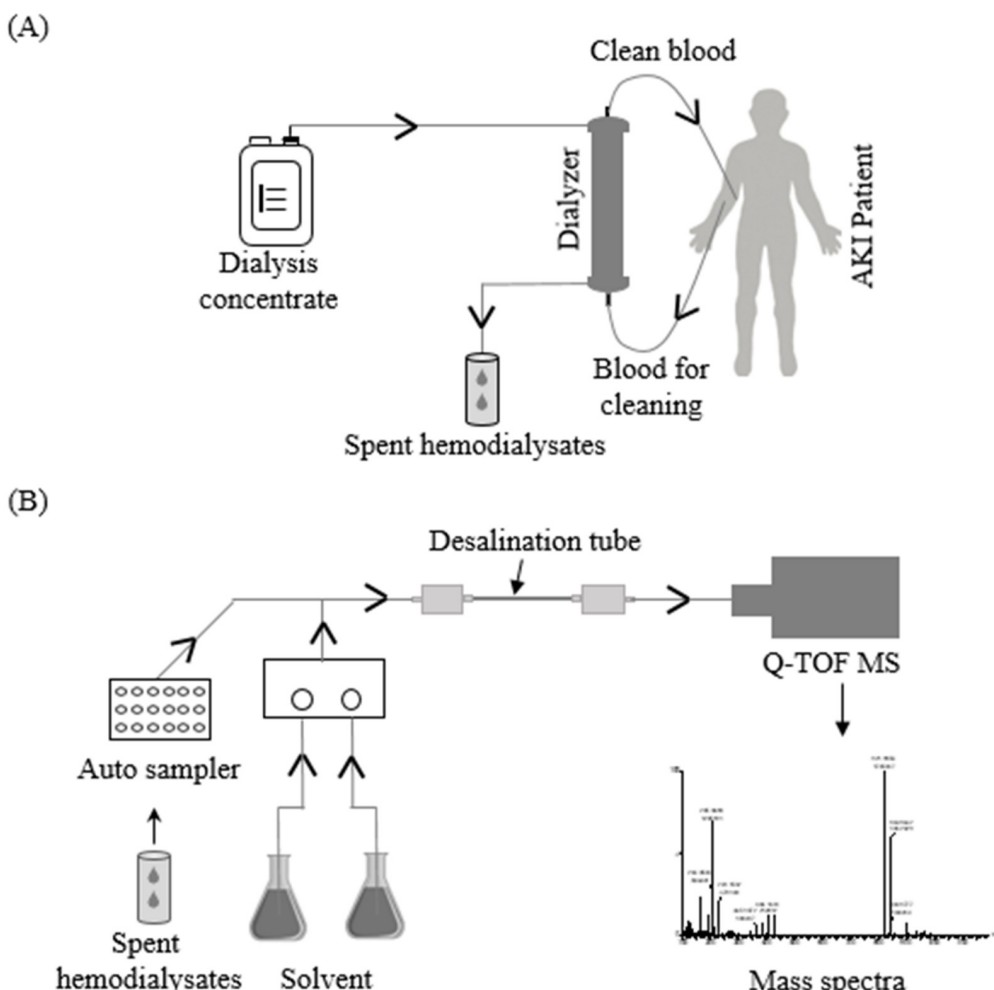

**Fig 1. An overview of spent hemodialysate analysis using online desalination tubes.** (A) spent hemodialysates were collected from acute kidney injury (AKI) patients and (B) subsequently analyzed by online desalination tube coupled with MS.

source (Fig 1B). It is capable of removing both cation and anion simultaneously by adsorbing excess salts needed to be purged from samples or mobile phase during analysis [13, 14].

Iodinated Contrast Media (ICM) are widely used for therapeutic and diagnostic imaging purposes [15]. However, the onset of new complications or exacerbation of renal dysfunction following the administration of ICM occurs among high-risk patients with preexisting chronic renal failure, diabetes, and multiple myeloma. For example, ICM showed nephrotoxic effects in patients with diabetes and impaired renal function undergoing coronary or aortofemoral angiography [16]. Their multiple exposures cause a direct cytotoxic effect on renal tubular epithelial and endothelial cells, resulting in impaired intrarenal hemodynamics, hypoxia, ischemia, and necrosis [17]. Therefore, the detection of ICM in spent hemodialysates is advantageous in monitoring the effect of continuous hemodiafiltration (CHDF) treatment. It will provide new indications about the appropriate dosages, route of administration, and comprehensive risk assessment versus the possible benefit of the contrast-assisted investigation or an alternative imaging strategy concerning the patient's clinical condition.

A variety of approaches have been applied in order to measure ICM, such as iohexol and ioversol in plasma [18], serum [19], urine [20], and wastewater [21], but no analytical method has been reported for the detection of ICM in spent hemodialysates matrix. Several approaches for online desalting and direct infusion of biological samples without chromatographic separations have been developed [22–24]. However, almost all of them have common weaknesses, mainly the ion suppression effect, due to the lack of chromatographic separations. Our simple and convenient approach enables the direct injection of untreated salty spent hemodialysates into MS and subsequently detects iohexol and ioversol, bringing it closer to its use in clinical practice. The online desalting approach offers speed and labor decrease for samples clean-up based on the ion-exchange technique resulting in faster performances than conventional analytical methods. This configuration is also beneficial for overcoming the ion suppression effect, which results in increased signal intensity and improved spectral signal to noise ratio (S/N). In the present work, we described the advantages of using the online desalting tube coupled to quadrupole time of flight (Q-TOF) mass spectrometer (Synapt G2 Q-TOF MS) for the detection of iodinated contrast agent iohexol and triple quadrupole (TQ) mass spectrometer (Xevo TQ-XS MS) for the detection of iodinated contrast agent ioversol in untreated spent hemodialysates collected from the patients with preexisting AKI without laborious sample preparation.

## Materials and methods

### Chemicals and equipment

Ioversol (Brand name: Optiray® 350) was purchased from Guerbet, Japan Co., Ltd. (Tokyo, Japan). LC-MS grade methanol, acetonitrile, formic acid, and water were purchased from Wako Pure Chemical Industries (Osaka, Japan). Ammonium acetate solution was purchased from Kanto Chemical Co., Inc. (Tokyo, Japan). CFAN desalination tubes (Lot no: 10100CFAN20001) were purchased from MS-Solutions (Tokyo, Japan).

### Patients and sample collection

Spent hemodialysates were obtained from three patients admitted to the intensive care unit (ICU) department, Hamamatsu University Hospital, Japan. For Patient #1, we collected the spent hemodialysates at 2h, 4h, 6h, and 24h after the start of CHDF. He underwent a contrast-enhanced computed tomography (CT) scan at 6h after the beginning of CHDF and was given 150 mL of iohexol (300 mg I/mL) intravenously for that purpose. For patients #2 and #3, we collected the spent hemodialysates sampled at 0h, 0.5h, 1h, 2h, 4h, 6h, and 24h after the start of CHDF and stored them at -80˚C until use. Patient #2 underwent coronary angiography (CAG) the day before starting CHDF, and 30 mL of ioversol (350 mg I/mL) was administered intravenously in less than 40 min. Patient #3 underwent 2-times CAG and administrated 70 mL and 100 mL of ioversol (350 mg I/mL) intravenously one day and two days prior to CHDF, respectively (S1 Table). CHDF was commenced on the next day of CAG and continued up to 24h in patients #2 and #3. The ethics committee of Hamamatsu University School of Medicine, Hamamatsu, Japan approved this research work (ethical approval number:19–169). Patients who were scheduled for spent hemodialysates collection provided written informed consent to use their sample in this study.

### Preparation of stock solution and spent hemodialysate samples

The ioversol stock solution was prepared by mixing Optiray® 350 in 100% water at concentrations of 741 mg/mL and stored at -20˚C for up to 3 months for further use. Later, the working standard solution was prepared in 50% methanol at 10 μg/mL concentrations by diluting the

stock standard solution and kept at 4˚C for up to 1 month for further use. The spent hemodialysates were thawed entirely at 4˚C. The bio-fluids gently vortexed for 30 seconds and collected the upper portion into the LC vials. Samples were maintained at 10˚C in the autosampler, and 5 μL of the samples were injected.

## Instrumentation and MS apparatus for flow injection analysis (FIA)

**FIA by Synapt G2 Q-TOF MS.**   The direct flow injection analysis of spent hemodialysates was performed using the online desalination tube connected with an ESI source equipped with a Q-TOF MS (Synapt G2 Q-TOF MS, Waters, Milford, MA, USA) in positive ion mode. The spray solvent (20 mM ammonium acetate in 50% methanol) was maintained at a flow rate of 0.2 mL/min using a solvent pump (ACQUITY UPLC Binary Solvent Manager, Waters, Milford, MA, USA). The ESI source conditions in positive ion mode were optimized using a capillary voltage of 4.0 kV, a cone voltage of 30 V, a source temperature of 150˚C, a desolvation temperature of 450˚C, a cone gas flow of 50 L/h, and a desolvation gas flow of 800 L/h. Ions from spent hemodialysates were obtained in a range of $m/z$ 100 to 1000. The mass spectra were calibrated using sodium formate solution (500 μM) in 2-propanol: water (90:10, v/v) prior to measurements. We used lock spray (leucine-enkephalin solution; $m/z$ 556.28) to obtain high mass accuracy.

**FIA by Xevo TQ-XS MS.**   The direct flow injection of spent hemodialysates was performed using the online desalination tube connected with an ESI source equipped with a TQ-MS (Xevo TQ-XS MS, Waters, Milford, MA, USA) in positive ion mode. The spray solvent (0.1% formic acid in 50% methanol) was maintained at a flow rate of 0.2 mL/min using a solvent pump (ACQUITY UPLC Binary Solvent Manager, Waters, Milford, MA, USA). We optimized the TQ-MS parameters for maximum sensitivity as follows, scan mode: multiple reaction monitoring (MRM), ionization mode: positive, capillary voltage: 3.0 kV, cone voltage: 30 V, source temperature of 150˚C, desolvation temperature: 500˚C, desolvation gas flow: 600 L/h, cone gas flow: 150 L/h, $N_2$ gas pressure: 7.0 bar.

## Data analysis

MassLynx (Waters, Milford, MA, USA; version 4.1) software was used for data acquisition and analysis. For the S/N ratio calculation, the maximum signal height above the mean noise was divided by the root mean square deviation from the mean noise. The obtained $m/z$ peaks were identified by referring to the human metabolome database (HMBD) and previous literature.

## Results

### Detection of ICM & endogenous metabolites in untreated spent hemodialysates

We injected the spent hemodialysates of patient #1 directly into Q-TOF MS via the online desalination tube (Fig 2). As expected, the peaks ($m/z$ 821.89 and $m/z$ 843.86) corresponding to the contrast agent iohexol were detected only in the sample obtained after 24h of initiating CHDF (Fig 2D). At the same time, we also detected the peaks of three endogenous metabolites ($m/z$ 114.07, 162.11, and 229.15 corresponding to creatinine, L-carnitine, and N, N, N-tri-methyl-L-alanyl-L-proline betaine (TMAP) respectively) in samples obtained after 2h, 4h, 6h, and 24h of CHDF respectively (Fig 2A–2D and Table 1). Furthermore, the peaks ($m/z$ 807.87 and $m/z$ 829.86) corresponding to ioversol along with three endogenous metabolites (creatinine, L-carnitine, and TMAP) were detected in the sample obtained from patients #2 and #3

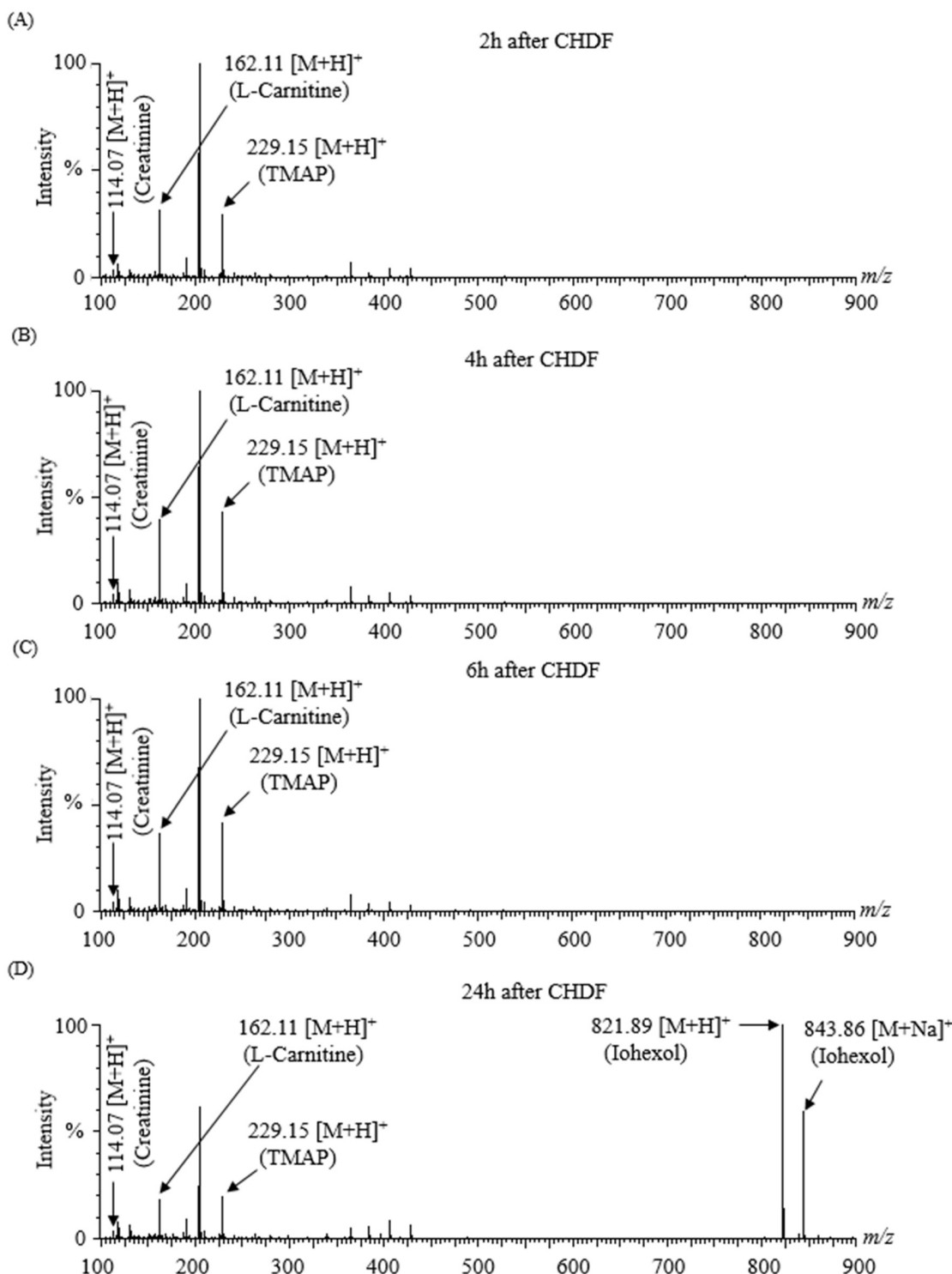

**Fig 2. Detection of iohexol (an ICM) and three endogenous metabolites in untreated spent hemodialysates of patient #1 by Synapt G2 Q-TOF MS.** The peaks corresponding to creatinine, L-carnitine, and TMAP were detected in samples obtained (A) 2h, (B) 4h, and (C) 6h after the start of CHDF. (D) Three endogenous metabolites along with the iohexol (*m/z* 821.89 and *m/z* 843.86) were detected in the sample collected 24h after the start of CHDF (iohexol was injected 6h after the start of CHDF).

**Table 1. List of peaks in the Q-TOF MS obtained from spent hemodialysates.**

| Observed $m/z$ | Theoretical $m/z$ | Assigned Molecules | Mass error (ppm) | Reference |
|---|---|---|---|---|
| 114.0665 | 114.0667 [M+H]$^+$ | Creatinine | 3 | [25] |
| 162.1105 | 161.1052 [M+H]$^+$ | L-carnitine | 12 | [2] |
| 229.1542 | 229.1552 [M+H]$^+$ | TMAP | 2 | [26] |
| 821.8853 | 821.8876 [M+H]$^+$ | Iohexol | 3 | [20] |
| 843.8632 | 843.8695 [M+Na]$^+$ | Iohexol | 7 | HMDB |
| 807.8700 | 807.8719 [M+H]$^+$ | Ioversol | 2 | [19] |
| 829.8620 | 829.8539 [M+Na]$^+$ | Ioversol | 10 | HMDB |

simultaneously (S1 Fig and Table 1). This technique can monitor the changes of metabolites with time (2-24h) in spent hemodialysates of AKI patients (S2 Fig).

## Evaluation of the ioversol adsorption within desalination tubes

We injected ioversol solution (10 μg/mL) into Q-TOF MS with or without the online desalination tubes to test the adsorption of ioversol within the tubes. We observed a stable peak for ioversol in both cases. When the ioversol solution was injected using the desalination tube, the order of adduct ions was dominated by [M+H]$^+$>[M+NH$_4$]$^+$>[M+Na]$^+$, with total intensity of $1.98 \times 10^4$. On the other hand, when the ioversol solution was injected without the desalination tube, the order of adduct ions was dominated by [M+H]$^+$>[M+Na]$^+$>[M+NH$_4$]$^+$, with total intensity of $1.46 \times 10^4$. It revealed almost similar signal intensity with the exception of the ratio of those adducts ions. This finding suggests that ioversol was not absorbed within the desalination tubes (Fig 3, Table 2).

## The effect of the online desalination tube in the measurement of untreated spent hemodialysates

We applied the online desalination tube to the mass spectrometry system to directly examine the untreated spent hemodialysates. This setup allows the direct injection of untreated spent hemodialysates into TQ-MS without additional sample preparation. We observed stable and higher signal sensitivity for ioversol due to its desalting ability. This approach enhances the signal intensity and improves the S/N for ioversol ($m/z$ 807.9>588.8) by approximately 178 times and 8 times, respectively, compared to data obtained without using it (Fig 4A and 4D, and S3 Table). In addition to increased sensitivity, the usage of online desalination tubes mitigates the peak broadening during analysis. By applying this method, we found the same product ion transition ($m/z$ 807.9>588.8) in untreated spent hemodialysates of patients #2 and #3 (S3 Fig). In addition, the lowest detectable concentration of ioversol is 0.1 ng/mL in spent hemodialysates or methanol, with S/N of approximately 4 or 33, respectively (S4 and S5 Figs).

## Tracking the changes of ioversol in spent hemodialysates with time

We examined the untreated spent hemodialysates of patients #2 and #3 at 7-time intervals (0-24h) following the initiation of CHDF. The online desalination tube coupled with TQ-MS can track the changes of ioversol in spent hemodialysates with time (0-24h). We observed higher ioversol signal intensity of patient #2 at 1h and subsequently decreased the signal intensity at 2h, 4h, 6h, and 24h after starting CHDF. Patient #3 had the higher ioversol signal intensity at the beginning (0h) and then decreased it after 0.5h, 1h, 2h, 4h, 6h, and 24h of starting CHDF. The ioversol signal intensity was consistently higher in patient #3 compared to patient #2 (Fig 5).

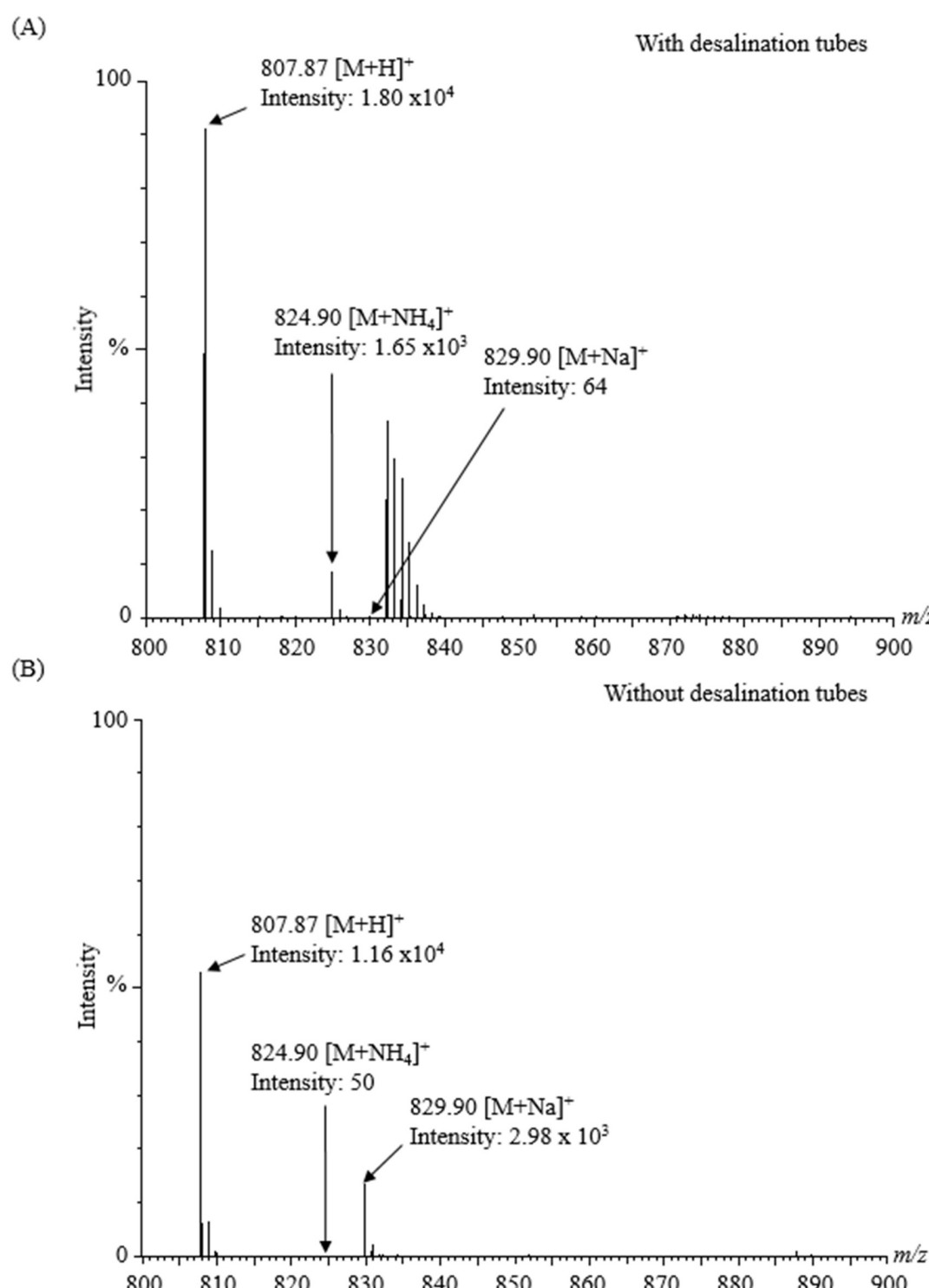

**Fig 3. Ioversol was not adsorbed by desalination tubes.** (A) Mass spectra were obtained using the desalination tube and (B) mass spectra were obtained without the desalination tube.

**Table 2. Q-TOF MS data for the evaluation of ioversol adsorption by desalination tube.**

| Status of desalination tube during analysis | Intensity values of ioversol | | | Total intensity |
|---|---|---|---|---|
| | $[M+H]^+$ | $[M+Na]^+$ | $[M+NH_4]^+$ | |
| With desalination tube | $1.80 \times 10^4$ | 64 | $1.65 \times 10^3$ | $1.98 \times 10^4$ |
| Without desalination tube | $1.16 \times 10^4$ | $2.98 \times 10^3$ | 50 | $1.46 \times 10^4$ |

## Discussion

In this study, we connected the online desalination tube before ESI-MS that achieved direct injection of untreated salty spent hemodialysates and simultaneously detected the ICM and endogenous metabolites. Previously, our group developed this desalination tube and showed

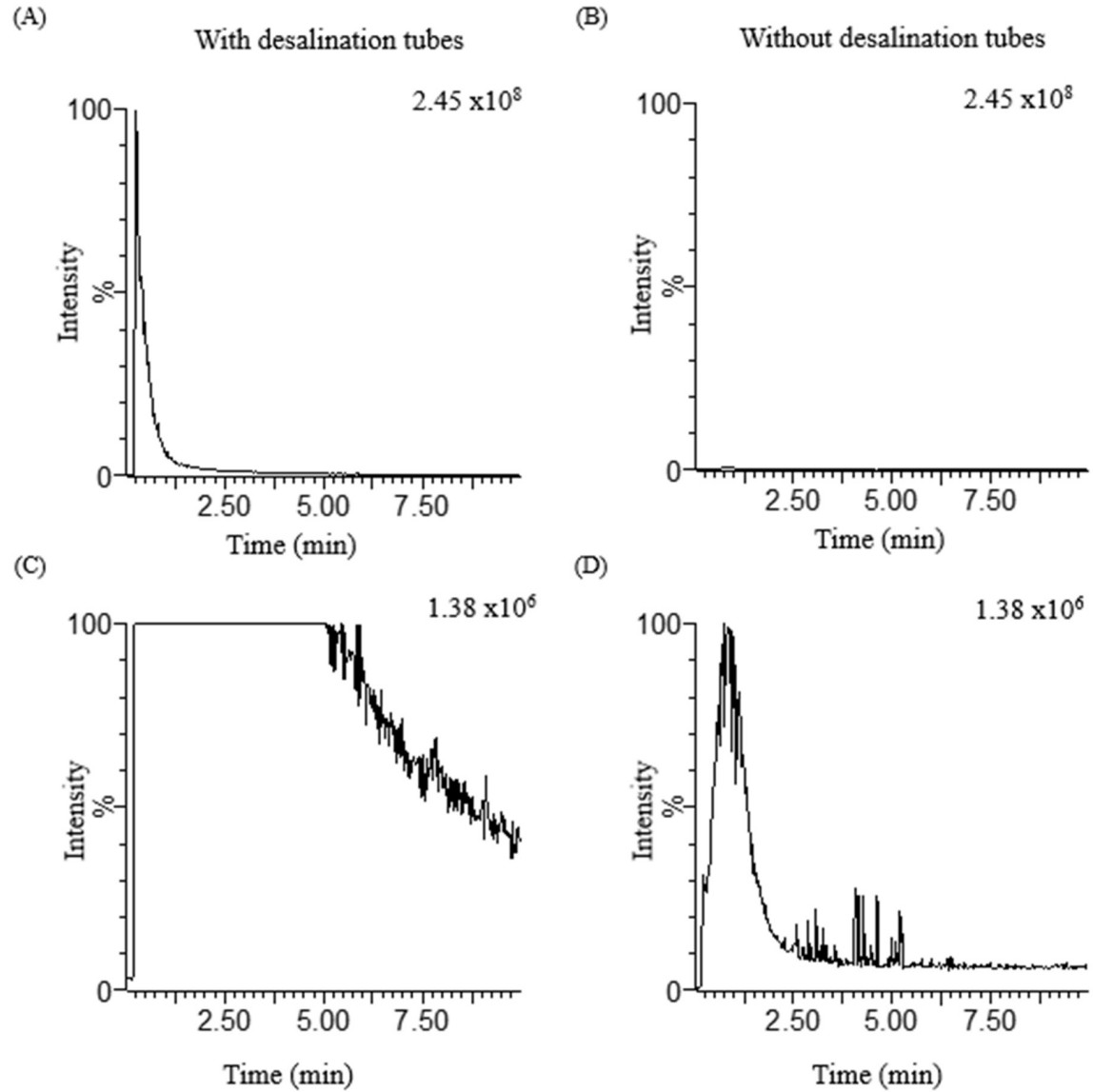

**Fig 4. Differences in the measurement results of untreated spent hemodialysates with and without an online desalination tube.**
When spent hemodialysates were injected into TQ-MS directly using (A, C) and without using (B, D) an online desalination tube, which revealed the differences in the measurement results. (A), (B) denoted the ioversol signal intensity in the same scale bar, and (C), (D) denoted the ioversol signal intensity in the same scale bar.

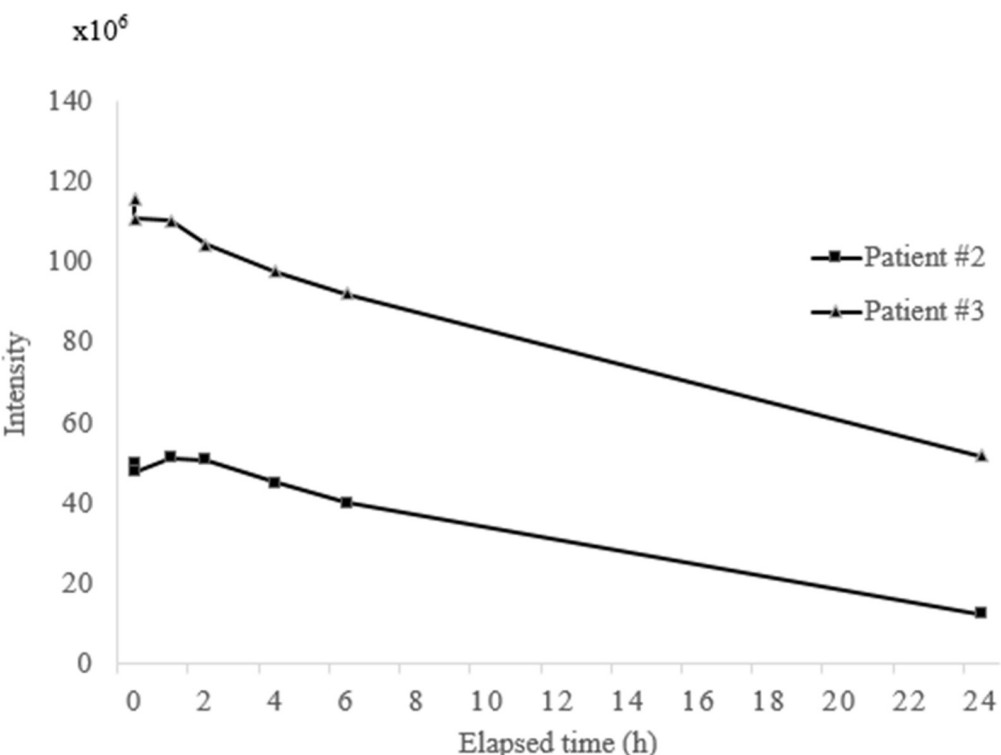

**Fig 5. The changes of ioversol signal intensity with time (0-24h) in spent hemodialysates.** The square shape and pyramid shape lines denote the time-dependent changes of ioversol signal intensity in patients #2 and #3, respectively.

its efficiency. The adsorption rate for phosphate and potassium in two distinct solutions (10 mmol/L $KH_2PO_4$ in 50% acetonitrile and 10 mmol/L $KH_2PO_4$ in 100% water) was more than 82% for up to 25 min, demonstrating its desalting ability [14]. This finding motivates us to apply the desalination tube to analyze salty, spent hemodialysates. It acts as a contaminant remover that causes the diffusion in the physical spaces within the tubes or adsorption on ion-exchange resin depending on the analytes prior to ESI-MS analysis [13]. Our study describes the benefit of using the online desalination tube coupled to Q-TOF MS that aids in detecting the ICM (iohexol) and other three endogenous metabolites concomitantly (Fig 2 and Table 1) in salty spent hemodialysates without sample extraction and chromatographic separations. It may purify the spent hemodialysates based on the ion-exchange technique, consequently resolving issues such as excess inorganic salts ($Na^+$, $K^+$, $Mg^{2+}$, $Ca^{2+}$, $Cl^-$, $SO_4^{2-}$, and $PO_4^{3-}$), nonvolatile solutes/buffers, and other impurities. We observed the identical endogenous metabolites in patients #2 and #3, with the exception of the iohexol. As expected, iohexol was found in patient #1, and ioversol was found in patients #2 and #3 (S1 Fig). This approach showed stable and higher signal sensitivity for contrast agents and other endogeneous metabolites concurrently due to the online desalting capability (Fig 2). After that, we checked the adsorption of ioversol within the desalination tubes by injecting ioversol solution (10 μg/mL ioversol in 50% methanol) into Q-TOF MS with or without the desalination tubes. The results showed a stable peak for ioversol and the adduct ions dominated by $[M+H]^+$ in both cases. The ratio of adduct ions detected differed depending on whether tubes were used or not (Fig 3), but the total intensity was higher with tubes (Table 2). This observation suggests that the adsorption of ioversol by the desalination tubes does not need to be considered.

MRM profiling was carried out using a TQ-MS with three aligned quadrupole masses for the salty spent hemodialysates study. To optimize the discrimination of transitions, we infused ioversol (10 μg/ml) into TQ-MS via an online desalination tube at a flow rate of 0.2 mL/min and interrogated using 7-different collision energy (CE). The product-ion transition (*m/z* 807.9>588.8) was optimized based on their best intensity by considering specific CE 25 eV (S2 Table). The MRM transition of ioversol (*m/z* 807.9>588.8) was organized into a method to measure the untreated spent hemodialysates directly. Thereafter, we examined the effect of online desalination tubes in measuring untreated salty spent hemodialysates. The application of an online desalination tube coupled with TQ-MS reduces ion suppression effects, resulting in increased signal intensity and improved S/N ratio (Fig 4A and 4C). In contrast, severe ion suppression and peak broadening were observed without the desalination tube, resulting in reduced signal intensity (Fig 4B and 4D). Our approach showed approximately 178 times higher signal intensity and 8 times greater S/N for the ioversol compared to those obtained without the desalination tube (Fig 4A and 4D, and S3 Table). The increase in signal intensity and the improvements in the S/N ratio significantly empower TQ-MS performance to examine the untreated salty spent hemodialysates directly without sample extraction and chromatographic separations. Using our method, we examined the untreated salty spent hemodialysates of patients #2 and #3 and obtained the specific ion pairs (*m/z* 807.9>588.8) within 1 min (S3 Fig), which does not require additional sample handling and chromatographic run time. To check the limit of detection (LOD), we spiked spent hemodialysates (collected before ioversol injection) or methanol with ioversol at 6 known concentration levels ranging from 0.01 to 1000 ng/mL. This method detected the lowest known concentration (0.1 ng/mL) of ioversol with the S/N of approximately 4 or 33 (S4 and S5 Figs). The conventional workflows used for detecting the ICM require sample extraction steps (protein precipitation, centrifugation, drying, dilution) with chromatographic separation [18–21]. Conversely, we detected the iohexol and ioversol in untreated spent hemodialysates directly without additional sample extraction and chromatographic separation. Furthermore, we were able to mitigate the ion suppression effects and acquired a stable signal for ICM and other endogenous metabolites, which is unusual during the direct examination of untreated biological samples [27].

Our study showed the benefit of employing the online desalination tube, which strengthens the TQ-MS to monitor the changes of ioversol in spent hemodialysates of AKI patients during CHDF. The ioversol signal intensity in patient #3 was higher in contrast to patient #2 because she had undergone 2-times CAG and received higher ioversol dosages before initiating CHDF (Fig 5). The higher dosages of ioversol may further worsen the renal complications in patients with types-2 diabetes, as an earlier study reported that ICM are exclusively associated with contrast-induced nephropathy in patients with diabetes mellitus [28]. Our approach showed the ability to tract ioversol changes with time in the AKI patient's spent hemodialysates. The changes in ioversol signal intensity were time-dependent (0-24h) in patients #2 and #3 (Fig 5), suggesting that CHDF removes ioversol with time. This phenomenon may reduce the severe risk factor for contrast-induced AKI and other complications. Although the elimination rate and half-life of ioversol determination demand quantitative analysis using blood samples, this technique represents the online noninvasive real-time monitoring of the changes of ioversol with time (0-24h) by measuring untreated spent hemodialysates.

## Conclusion

The online desalination tube coupled with MS showed the capability of direct detection of iohexol and ioversol in untreated salty spent hemodialysates with high sensitivity. It also

showed the ability to track the changes of ioversol in spent hemodialysates of AKI patients during CHDF. It is expected to be applied in evaluating the clinical predictors of contrast-induced AKI in the future.

## Supporting information

**S1 Fig. Detection of ICM and three endogenous metabolites in untreated spent hemodialysates of patients #2 and #3 by Synapt G2 Q-TOF MS.** (A) and (B), Three endogenous metabolites along with the ioversol ($m/z$ 807.87 and $m/z$ 829.86) were detected in untreated spent hemodialysates of patients #2 and #3 at the beginning of CHDF.
(DOCX)

**S2 Fig. The changes of metabolites with time (2-24h) in spent hemodialysis of patient #1 during CHDF.**
(DOCX)

**S3 Fig. Detection of ioversol in untreated spent hemodialysates by MRM profiling.** Patients #2 and #3 were found ioversol positive, denoted as (A) and (B), respectively.
(DOCX)

**S4 Fig. Determination of the limit of detection of ioversol in methanol (A) or spent hemodialysates (B).** The left top denoted blank (methanol), whereas the right top denoted blank (spent hemodialysates), Later, we spiked with known amounts of ioversol ranging from 0.01 to 1000 ng/mL. Here, the lowest detectable concentration of ioversol is 0.1 ng/mL in spent hemodialysates or methanol.
(DOCX)

**S5 Fig. The signal to noise ratio (S/N) of ioversol in methanol (A) and spent hemodialysates (B).**
(DOCX)

**S6 Fig. Chemical structure of the metabolites, ICM (iohexol, ioversol), and ioversol product detected in spent hemodialysates.**
(DOCX)

**S1 Table. Patient characteristics.**
(DOCX)

**S2 Table. Detailed MRM transition and compound-dependent parameters.**
(DOCX)

**S3 Table. Comparison of the signal intensity and spectral signal-to-noise ratio (S/N) of ioversol with and without an online desalination tube.**
(DOCX)

**S4 Table. The signal intensity of ioversol ($m/z$ 807.9) with time (0-24h) in spent hemodialysates of patients #2 and #3.** (Numerical intensity value of Fig 5).
(DOCX)

**S5 Table. The signal intensity of metabolites ($m/z$ 162.11, $m/z$ 229.15, $m/z$ 114.07, $m/z$ 203.05, and $m/z$ 205.07) with time (2-24h) in spent hemodialysates of patient #1.** (Numerical intensity value of S2 Fig).
(DOCX)

## Acknowledgments

We would like to thank the medical engineer Suzuki and the supporting staff at the Blood Purification Therapy Department, Hamamatsu University School of Medicine (HUSM).

## Author Contributions

**Conceptualization:** Akihiko Kato, Mitsutoshi Setou, Tomohito Sato.

**Data curation:** Md. Mahamodun Nabi, Takumi Sakamoto, Ariful Islam.

**Formal analysis:** Md. Mahamodun Nabi, Md. Al Mamun, Ariful Islam.

**Funding acquisition:** Akihiko Kato, Mitsutoshi Setou, Tomohito Sato.

**Investigation:** Md. Mahamodun Nabi, Takumi Sakamoto, Md. Al Mamun, Ariful Islam, Mitsutoshi Setou, Tomohito Sato.

**Methodology:** Md. Mahamodun Nabi, Tomoaki Kahyo, Mitsutoshi Setou, Tomohito Sato.

**Project administration:** Akihiko Kato, Mitsutoshi Setou, Tomohito Sato.

**Resources:** Shuhei Aramaki, Shingo Ema, Akihiko Kato, Yutaka Takahashi, Mitsutoshi Setou.

**Supervision:** Yutaka Takahashi, Tomoaki Kahyo, Mitsutoshi Setou, Tomohito Sato.

**Writing – original draft:** Md. Mahamodun Nabi, Takumi Sakamoto, Md. Al Mamun, Ariful Islam, A. S. M. Waliullah, Tomohito Sato.

**Writing – review & editing:** Md. Mahamodun Nabi, Takumi Sakamoto, Md. Al Mamun, Ariful Islam, A. S. M. Waliullah, Shuhei Aramaki, Md. Mahmudul Hasan, Tomoaki Kahyo, Tomohito Sato.

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
