## [Decision Letter · Decision Letter 0]

4 Mar 2022

PONE-D-22-03854A convenient online desalination tube coupled with mass spectrometry for the direct detection of iodinated contrast media in untreated human spent hemodialysatesPLOS ONE

Dear Dr. Sato,

Thank you for submitting your manuscript to PLOS ONE. After careful consideration, we feel that it has merit but does not fully meet PLOS ONE’s publication criteria as it currently stands. Therefore, we invite you to submit a revised version of the manuscript that addresses the points raised during the review process.

We look forward to receiving your revised manuscript.

Kind regards,

Joseph Banoub, Ph,D., D. Sc., FCIC, FRSC

Academic Editor

PLOS ONE

Journal Requirements:

Additional Editor Comments:

This manuscript needs major corrections that need to be worked out in your revised form.

Reviewers' comments:

Reviewer's Responses to Questions

**Comments to the Author**

1. Is the manuscript technically sound, and do the data support the conclusions?

Reviewer #1: Yes

Reviewer #2: Yes

2. Has the statistical analysis been performed appropriately and rigorously? 

Reviewer #1: Yes

Reviewer #2: N/A

3. Have the authors made all data underlying the findings in their manuscript fully available?

Reviewer #1: Yes

Reviewer #2: Yes

4. Is the manuscript presented in an intelligible fashion and written in standard English?

Reviewer #1: Yes

Reviewer #2: Yes

5. Review Comments to the Author

Reviewer #1: This manuscript present a very interesting work using Mass sprectrometry for the detection of biomomolecules in body fluids.

First of all, as chemist, I think it would be interesting to report not only the structures of the molecules detected in MS and indicated in table 1, but also of the ioversol and especially at least a proposal for the structure of the products ion at m/z 588.8 .

Please write correctly the formula of sulfate and phospate ions (Line 270)

Authors are also asked to pay attention to the visual quality of the figures, as they are blurry and sometimes difficult to read

Finnaly please check the style of the references so that they match to the guideline for authors corresponding to the journal

Reviewer #2: This manuscript presents an interesting application using a desalination tube fitted online to the mass spectrometer to analyze directly untreated human spent hemodialysates for the presence of hemodialisates. The technique used for sample clean-up is based on ion exchange; the concept of inline sample clean up prior to LC-MS of biomolecular solutions dates back to 2003 (Ref 5) and there is a report using a desalination tube for online LC-MS analysis published in 2021 (Ref 12).

This manuscript is well prepared, with current references and the findings are very interesting. The introduction and approach to sample preparation are thorough.

Please find below a couple of comments, questions and concerns:

1. In figures 2 and 3, the Intensity is expressed in %units. Regarding Patient #1: how do you know that CHDF “consistently removed the metabolites such as creatinine, L-cartinine, TMAP and iohexol with time”(lines 207-208)? Were you using an internal standard? It is difficult to estimate a decreasing trend when the Y-axis units are expressed in %.

2. Iohexol was not detected in patient #1 until after 24h of initiating CHDF (line 201). In patients #2 and #3 ioversol is detected in a higher concentration in the first couple of hours; ioversol signal decreases over time with the lowest concentration after 24h (lines 238-242).

Both iohexol and ioversol are contrasting agents. Why would you see such an opposite behaviour and what does it say about the use of these two contrasting agents?

3. In Patient #1, a range of metabolites such as creatinine, L-carnitine, TMAP are detected (lines 202-204). Were these metabolites observed in Patients #2 and #3? Can you comment on their signal levels with increase in time after the start of CHDF in Patients #2 and #3? Do these trends compare to what was observed in Patient #1?

4. What is the desalination efficiency of these tubes in terms of removing the excess inorganic salts ? Are there any reference studies that address this question?

5. Ioversol standards were infused via the online desalination tube into the TW MS/MS for method optimization. Can you comment if there was any loss of ioversol during the desalination tube stage? Were you able to establish a concentration range for the ioversol signal detected in the different samples?

6. How does the signal intensity for ioversol or iohexol help with determining and establishing the maximum and/ or safe levels of these contrasting agents in hemodialysates? Is there a known limit of detection for ioversol or iohexol in hemodialysates when using these instruments?

7. Lines 256-267, 277-279, 304-314 in the Discussion section contain review type’ information that would best fit in the Introduction section.

Although the findings are important for this application, there is not enough data for a full research article. So I suggest resubmitting as a communications article or major revision to add more data and address most concerns expressed here.

6. PLOS authors have the option to publish the peer review history of their article (what does this mean?). If published, this will include your full peer review and any attached files.

Reviewer #1: No

Reviewer #2: No

---

## [Author Response · Author response to Decision Letter 0]

28 Apr 2022

Response to Reviewer #1

Comment #1: This manuscript present a very interesting work using Mass sprectrometry for the detection of biomomolecules in body fluids. First of all, as chemist, I think it would be interesting to report not only the structures of the molecules detected in MS and indicated in table 1, but also of the ioversol and especially at least a proposal for the structure of the products ion at m/z 588.8 .Please write correctly the formula of sulfate and phospate ions (Line 270) SO42-, PO43-, Authors are also asked to pay attention to the visual quality of the figures, as they are blurry and sometimes difficult to read

Finnaly please check the style of the references so that they match to the guideline for authors corresponding to the journal

Author response to comment #1: Thank you for the comment and suggestion. We have corrected the formula of sulfate (SO42–) and phosphate ions (PO43–) (Page:15, Line 279-280). The chemical structure of all compounds detected in spent hemodialysates has been provided as a supplementary figure (S6 Fig). The visual quality of the figures has been improved, and the reference style has been corrected accordingly.

Response to Reviewer #2

Comment #1: This manuscript presents an interesting application using a desalination tube fitted online to the mass spectrometer to analyze directly untreated human spent hemodialysates for the presence of hemodialisates. The technique used for sample clean-up is based on ion exchange; the concept of inline sample clean up prior to LC-MS of biomolecular solutions dates back to 2003 (Ref 5) and there is a report using a desalination tube for online LC-MS analysis published in 2021 (Ref 12).

This manuscript is well prepared, with current references and the findings are very interesting. The introduction and approach to sample preparation are thorough.

Please find below a couple of comments, questions and concerns:

1. In figures 2 and 3, the intensity is expressed in %units. Regarding Patient #1: how do you know that CHDF "consistently removed the metabolites such as creatinine, L-cartinine, TMAP and iohexol with time" (lines 207-208)? Were you using an internal standard? It is difficult to estimate a decreasing trend when the Y-axis units are expressed in %.

Author response to comment #1: Thank you for the comment and suggestion. We agree that it is difficult to estimate a decreasing trend when the intensity is expressed in % units. We did not use internal standards. Therefore, we revised that line as follows "This technique can monitor the changes of metabolites with time (2-24h) in spent hemodialysates of AKI patients." (Page:11, Line: 210-211). This data is also included in the supplementary figure (S2 Fig).

Comment #2: Iohexol was not detected in patient #1 until after 24h of initiating CHDF (line 201). In patients #2 and #3 ioversol is detected in a higher concentration in the first couple of hours; ioversol signal decreases over time with the lowest concentration after 24h (lines 238-242).

Both iohexol and ioversol are contrasting agents. Why would you see such an opposite behaviour and what does it say about the use of these two contrasting agents?

Author response to comment #2: Thank you for the comment. Actually, the time interval (0h, 0.5h,1h, 2h, 4h, 6h, and 24h) shown in the manuscript was based on the beginning of CHDF, not on the ICM injection. May be the information regarding the ICM injection, and sample collection was not clear in the manuscript. We would like to clarify this issue. 

In patient #1, the iohexol was given 6h after the CHDF beginning. We collected the spent hemodialysates at 2h, 4h, 6h, and 24h after starting CHDF. As expected, we detected iohexol in spent hemodialysates of this patient after 24h. 

In patient #2, the ioversol was given the day before initiating CHDF. We collected the spent hemodialysates at 0h,0.5h, 1h, 2h, 4h, 6h, and 24h after starting CHDF. Accordingly, we detected ioversol in spent hemodialysates of this patient from the beginning (0h). 

In patient #3, The ioversol was given one day and two days before initiating CHDF. We collected the spent hemodialysates at 0h, 0.5h, 1h, 2h, 4h, 6h, and 24h after starting CHDF. As a result, we detected ioversol in spent hemodialysates of this patient from the beginning (0h). We have included the information on sample collection in the materials and methods section (Page:8, Line:145-154).

Comment #3: In Patient #1, a range of metabolites such as creatinine, L-carnitine, TMAP are detected (lines 202-204). Were these metabolites observed in Patients #2 and #3? Can you comment on their signal levels with an increase in time after the start of CHDF in Patients #2 and #3? Do these trends compare to what was observed in Patient #1?

Author response to comment #3: Thank you for your comment and suggestion. Patients #2 and #3 had the same metabolites (creatinine, L-carnitine, and TMAP) as patient #1, with the exception of the contrast agent iohexol. As expected, iohexol was found in patient #1, and ioversol was found in patients #2 and #3 (Page:11, Line: 202; 208 Page:15, Line: 282). We added this information as the supplementary figure (S1 Fig). We could not compare the metabolite trends in all patients because the samples were not collected at the same time periods. 

Comment #4: What is the desalination efficiency of these tubes in terms of removing the excess inorganic salts? Are there any reference studies that address this question?

Author response to comment #4: Thank you for your comment and suggestion. Previously, it has been shown that the adsorption rate of the desalination tube for phosphate and potassium in two different solutions (10 mmol/L KH2PO4 in 50% acetonitrile and 10 mmol/L KH2PO4 in 100% water) was more than 82% for up to 25 min (Ref #14). We have included the information in the discussion part (Page:14-15, Lines: 268-271). 

Comment #5: Ioversol standards were infused via the online desalination tube into the TW MS/MS for method optimization. Can you comment if there was any loss of ioversol during the desalination tube stage? Were you able to establish a concentration range for the ioversol signal detected in the different samples?

Author response to comment #5: Thank you for your comment. It is hard to measure the loss of ioversol from the MRM data of TQ-MS as it shows the chromatogram of a specific adduct. Alternatively, we checked the loss of ioversol using Q-TOF MS, and the data is shown in Table 2 (Page:13). Also, we established a concentration range for ioversol by spiking in methanol or spent hemodialysates (collected before ioversol injection) using six known concentrations of ioversol ranging from 0.01 to 1000 ng/mL. We have included these data in the supplementary figures (S4 Fig).

Comment #6: How does the signal intensity for ioversol or iohexol help with determining and establishing the maximum and/ or safe levels of these contrasting agents in hemodialysates? Is there a known limit of detection for ioversol or iohexol in hemodialysates when using these instruments?

Author response to comment #6: Thank you for your comments. To the best of our knowledge, there is no known limit of detection for ioversol or iohexol in spent hemodialysates. To check the limit of detection (LOD), we spiked spent hemodialysates (collected before ioversol injection) or methanol using ioversol at six concentration levels ranging from 0.01 to 1000 ng/mL. The LOD of the current method is 0.1 ng/mL for tested ioversol in spent hemodialysates or methanol (S4 Fig). We checked the concentration of ioversol (0.1 ng/mL) by spiking in spent hemodialysis or methanol of 7 independent replicates, ensuring the concentration fulfilled 95% of the consistency. The signal-to-noise ratio (S/N) of the lowest known concentration of ioversol (0.1 ng/mL) is approximately 4 or 33 in spent hemodialysates or methanol. We have added this information in the result part (Page:13; Line: 245-246) and supplementary figure (S5 Fig).

Comment #7: Lines 256-267, 277-279, 304-314 in the Discussion section contain review type' information that would best fit in the Introduction section. Although the findings are important for this application, there is not enough data for a full research article. So I suggest resubmitting as a communications article or major revision to add more data and address most concerns expressed here.

Author response to comment #7: Thank you for your comment and suggestion. We have moved some parts of those sentences into the introduction part, and the rest of those sentences have been deleted from the revised version (Page: 5; Line: 78).

---

## [Editor Report · Decision Letter 1]

9 May 2022

A convenient online desalination tube coupled with mass spectrometry for the direct detection of iodinated contrast media in untreated human spent hemodialysates

PONE-D-22-03854R1

Dear Dr. Sato,

We’re pleased to inform you that your manuscript has been judged scientifically suitable for publication and will be formally accepted for publication once it meets all outstanding technical requirements.

Kind regards,

Joseph Banoub, Ph,D., D. Sc., FCIC, FRSC

Academic Editor

PLOS ONE
---

## [Editor Report · Acceptance letter]

27 May 2022

PONE-D-22-03854R1 

A convenient online desalination tube coupled with mass spectrometry for the direct detection of iodinated contrast media in untreated human spent hemodialysates 

Dear Dr. Sato:

I'm pleased to inform you that your manuscript has been deemed suitable for publication in PLOS ONE. Congratulations! Your manuscript is now with our production department. 

Kind regards, 

on behalf of

Dr. Joseph Banoub 

Academic Editor

PLOS ONE